# Assessment of Commercial Fungicides against Onion (*Allium cepa*) Basal Rot Disease Caused by *Fusarium oxysporum* f. sp. *cepae* and *Fusarium acutatum*

**DOI:** 10.3390/jof7030235

**Published:** 2021-03-21

**Authors:** Ofir Degani, Ben Kalman

**Affiliations:** 1Plant Sciences Department, MIGAL—Galilee Research Institute, 2 Tarshish St., Kiryat Shmona 11016, Israel; kalman35@gmail.com; 2Faculty of Sciences, Tel-Hai College, Upper Galilee, Tel-Hai 12210, Israel; 3The Mina and Everard Goodman Faculty of Life Sciences, Bar-Ilan University, Ramat Gan 52900, Israel

**Keywords:** *Allium cepa*, basal rot, chemical control, fungus, *Fusarium*, onion, pathogenicity assay, Prochloraz, seed infection

## Abstract

The onion basal rot disease is a worldwide threat caused by species of the genus *Fusarium.* Today, Israel’s control of this disease is limited to a four-year growth cycle and Metam sodium soil disinfection. Here, commercial chemical fungicides were evaluated as control treatments against two of the primary pathogens involved, *F. oxysporum* f. sp. *cepae* and *F. Acutatum*. Out of 10 fungicides tested on culture plates, 3, Prochloraz, Azoxystrobin + Tebuconazole, and Fludioxonil + Sedaxen, had strong inhibitory effects on mycelial growth and were selected and tested in seeds in vitro. The preparations were applied as a seed coating and tested in two commercial cultivars, Riverside (Orlando, white cv.) and Noam (red cv.). Prochloraz (0.3% *w*/*w* concentration), the most promising compound, was efficient in reducing the Noam cv. sprouts’ disease symptoms. This preparation had no harmful in situ-toxicity effect and did not influence the plants’ seed germination and early development. In Noam cv. potted 30-day-old sprouts, the Prochloraz treatment was able to reduce the harmful impact of *F. oxysporum* f. sp. *cepae*. on the seedlings’ wet biomass, but was not effective in the Riverside cv. or against the *F. acutatum* pathogen. This suggests that future protective strategies must include an effective protective suit tailored to each of the pathogen species involved and the onion cultivar. The methods presented in this work can be applied for rapidly scanning multiple compounds while gradually ruling out ineffective ones. Eventually, this screening will enable field testing of the highest potential fungicides that successfully pass the pot experiments.

## 1. Introduction

Onion *Allium cepa* L. is a member of the *Amaryllidaceae* family known as the common onion or bulb onion. Onion is an important vegetable crop globally and accounts for 23.8% of the world’s total vegetable production area (FAOSTAT, 2019 Food and Agriculture Commodity Production data). In Turkey, the world’s largest onion producer, onion *Fusarium* basal rot disease (FBR) is considered a significant threat to this cultivar. For instance, in Amasya, a prominent Turkish onion production region, more than 80% of onion fields are infected [1]. In India, which ranks second in global onion production, FBR is one of the most devastating onion diseases, and yield losses in the field and post-harvest storage can reach 50% [2]. In Israel, the cultivation area of dry onions was 4,024 hectares in 2019 (FAOSTAT, 2019). In 2017 in Israel, onion bulbs’ commercial production for local markets reached ca. 47,000 tons (9% of bulk fresh vegetables). The leading summer onion variety is Riverside (white Orlando cv.), which accounts for about 45% of the dry onion area (other summer varieties are Mars cv. and Noam cv.) and is grown in all parts of the country.

For many years, FBR has been known to be one of the most harmful diseases in Israel’s onion cultivation [3,4]. In recent years, the disease prevalence has increased in onion fields throughout Israel, especially among summer onion varieties (data according to the Israeli Ministry of Agriculture and Rural Development, Consultation Service (Shaham), Beit-Dagan). The reasons for the increase in the disease’s incidence may be higher temperatures and long-term continuous cropping [5]. Such damages were reported recently in northeastern Israel’s Golan Heights [3]. The exact loss of marketable bulbs in Israel due to the disease is unknown. The extent of infected crops in the growing area can reach 1%, but losses can extend beyond the field (data according to the Israeli Ministry of Agriculture and Rural Development). Infected onions do not always show disease symptoms, and if they arrive at storage facilities, the problem is considerably worsened [3]. When stored in open sheds or packing houses, the disease can quickly spread to other onion bulbs. Furthermore, there is a fear that infected onions without visual symptoms will reach markets across the country. This concern is strengthening since known toxins are produced by this pathogen genius [6].

In a recent study, four species were isolated in northeastern Israel (Golan Heights) from onion samples from infected fields, identified and characterized: *F. proliferatum*; *F. oxysporum* f. sp. *cepae*; and two lesser-known species as FBR agents, *F. acutatum* and *F. anthophilium* [3]. Still, other pathogenic *Fusarium* species may be involved in FBR, and significant knowledge gaps exist in Israel regarding the disease’s nature and distribution and the control methods applied against it. Specifically, no orderly information is available about the disease’s history, its spread over the years, and its current precise distribution map. Moreover, so far, no FBR-resistant onion cultivars have been identified, and no fungicides against the causal pathogens involved have been proven to be unequivocally effective.

The FBR disease’s primary cause is *Fusarium oxysporum* f. sp. *cepae* [7]. This fungus is a significant soil-borne disease that is prevalent worldwide in onion-growing areas, causing severe yield losses in the field and onion bulbs’ infection under storage conditions [5]. The disease occurs in all stages of the crop’s growth. Basal rot that is generally initiated from the inoculum present in the soil is more difficult to control since the symptoms appear when most of the damage has already been done. *Fusarium* spp. infection causes pre- and post-emergence damping-off, root rot of older plants, and steam plate discoloration and basal rot of bulbs in the field and in storage [8]. The infection symptoms at the basal stem portion of the onion bulb spread and may eventually kill the host plant [9]. The disease’s symptoms in the field are usually localized in patches. Over time, the spread of the disease increases in infected areas, and there are more spotty appearances in the field and loss of growing regions.

Today, in Turkey, no resistance sources against the disease have been identified nor developed [10]. Instead, the disease is controlled by chemical seed treatments with Antracol (surface protectants, broad-spectrum dithiocarbamates) and Carbendazim (systemic, broad-spectrum benzimidazole fungicide) [10]. However, by using synthetic agrochemicals with only one target site, resistance can quickly become a severe problem [11]. In India, FBR management through chemicals and resistant cultivars are efficient to some extent [2]. New studies suggest biological control of the disease as an important component of integrated disease management in both countries [1,2]. Still, if the soil is heavily infected, the measures known to minimize the crop losses brought about by disease, such as the use of fungicides, are limited.

The current study aimed at examining the potential of new chemical fungicides against two of the primary pathogens involved, *F. oxysporum* f. sp. *cepae* and *F. Acutatum*. The experiments include rapid plate assay screening of fungicides with different action mechanisms, an in vitro seed infection assay after seed treatment with selected fungicides, and eventually a chemical control pot experiment with the most promising compound identified by the previous tests.

## 2. Materials and Methods

### 2.1. Fungal Strains

Two *Fusarium* isolates, *F. acutatum* and *F. oxysporum* f. sp. *cepae*, were examined in this study (Table 1). Cultures were generated from single conidia to create colonies having a clean genetic source for further research. The isolates were identified as described in a previous study [3] based on colony structures and macroscopic characterization, and molecular analysis.

### 2.2. Plate Assay

The effect of selected fungicides in inhibiting the fungus *F. oxysporum* f. sp. *cepae* and *F. acutatum* was studied in culture plates. Here, Petri dishes (9 mm) containing PDA were prepared to test various fungicides (Table 2). The active ingredient concentration of each fungicide was set to 1, 10, and 100 PPM. Each concentration was tested in six repetitions, and the whole experiment was repeated twice with similar results. In addition, control plates were prepared with a PDA substrate (without the addition of pesticide). A 6-mm-diameter disc from the margin of a 4–6-day-old fungal colony (that was grown in the dark at 28 ± 1 °C) was placed at the center of each plate, and the plates were incubated at the above temperature in the dark. After six days, the diameter of the two fungi species in each growth medium was measured. The inhibition percentage caused by each anti-fungal compound was calculated.

### 2.3. Seed Germination Pathogenicity Assay

The seed pathogenicity test was designed to measure the effectiveness of selected fungicides in restricting the virulence of the two *Fusarium* spp. on onion seeds. Germination and the first developmental stage of the infected seeds were monitored under the influence of the fungicides. The assay was conducted in six replications, and the whole experiment was repeated twice with similar results. The Riverside (Orlando) cv. (white onion) and Noam cv. (red onion) seeds (supplied by Hazera Seeds Ltd., Berurim M.P. Shikmim, Israel) were tested. The seeds were rinsed in double-distilled water (DDW), soaked in 1% NaOCl for 1 min, and rinsed again twice in DDW. The seeds were coated by complete adsorption of the active substance into the seed by soaking for half an hour in fungicides in a sterile plastic bag according to the supplier’s (Gadot Agro, Israel) instructions. The concentrations (*w*/*w*) of the fungicides were according to the manufacturer’s instructions: Azoxystrobin + Tebuconazole (Azimut, Adama Makhteshim, Be’er Sheva, Israel)—33%, Prochloraz (Sportak, Gadot Agro, Kidron, Israel)—0.3%, and Fludioxonil + Sedaxen (Vibrance, Gadot Agro, Kidron, Israel)—20%.

Each group of 10 onion seeds was transferred to a Petri dish in which sterile Watman paper was soaked in water and inoculated with a 6-mm-diameter disc. The colony disc was cut from the margins of a 5-day-old colony of the two *Fusarium* isolates, *F. acutatum* and *F. oxysporum* f. sp. *cepae* (see Table 1). The *Fusarium* colonies were grown previously on PDA medium under dark conditions at 28 ± 1 °C. In the control group, a sterile 6-mm-diameter PDA disc was added to the seeds. Sterile tap water was added to each plate every three days to ensure moisture conditions and allow efficient germination and disease progression. The inspected seeds were grown for nine days under the above conditions. At the end of the experiment, the seeds were washed, and their germination percentages, sprout biomass, and epicotyl length were measured.

### 2.4. Pot Pathogenicity Assay

Prochloraz pesticide (brand name Sportak, Gadot Agro, Kidron, Israel) was applied in a seedlings pathogenicity assay in pots to protect onion plants against FBR. In the experiment, 0.5-L pots were used with a non-sterile commercial garden soil mixture (Garden Mix, Deshanit, Be’er Yaakov, Israel) composed of coconut peat, fibers, a relatively low amount of tuff, and Osmocote (ScottsMiracle-Gro, Marysville, Ohio, United States), a 3–4 month slow-release fertilizer. To each pot, four onion seeds of the Noam cv. were planted at a depth of 2 cm. The first inoculation was done after the seeds’ full emergence (five days from sowing) by adding two fungal discs (grown on a PDA substrate for six days at 28 °C in the dark) to each seed. The second inoculation was done four days after the first treatment by adding 2 mL of a spore suspension at a concentration of 10^6^ spores/mL to each plant near the root. The spore suspension was prepared by gently scraping the colony mycelium from the Petri dish surface using 10 mL of sterilized DDW and a Dargalski stick, and filtering through a sterile gauze pad for a suspension containing only propagation units without mycelium. About a week after the second inoculation (14 days from sowing), the anti-fungal compound treatment was carried out by irrigating with 100 mL tap water per pot. The fungicide Prochloraz concentration, 0.3% *w*/*w*, was chosen according to the manufacturer’s recommendation. The treatments without the pesticide were watered with 100 mL of tap water. The plants were grown in a growing room in an artificial light regime of 16 h and 8 hours of darkness, with 45–50% humidity at 28 ± 3 °C. The experiment lasted 30 days and ended after about two weeks from the day of pest control. The plant’s symptoms were documented at the experiment end, and wet biomass and stalk length were measured.

### 2.5. Statistical Analyses

A statistical design with full randomization was used in all the assays. This regards the Petri dishes’ positioning in the incubator and the seedlings pots in the growing room. Data analysis followed by statistics was done using the JMP program, 15th edition, SAS Institute Inc., Cary, NC, USA. The one-way analysis of variances (ANOVA) tracked by multiple comparisons post hoc of the Student’s *t*-test for each pair (without multiple comparisons correction) was used to estimate the differences at a significance level of *p* < 0.05.

## 3. Results

### 3.1. Plate Assay

Our research strategy was to estimate the efficiency of selected fungicides in three series of experiments that would gradually increase the investment in time and work without wasting effort if the treatment had not successfully passed the early steps. The first stage was a rapid screening of a relatively large number of commercial preparations in growth medium plates. Scanning and examination of fungicides in Petri dishes on a PDA substrate showed that most of them caused a significant (*p* < 0.05) decrease in *Fusarium* spp. isolates growth (Figure 1, Figure 2 and Figure 3). Four of these fungicides, Tebuconazole alone (Orius 25) or in mixture with Azoxystrobin (Azimut), Prochloraz (Sportak), and Fludioxonil + Sedaxen (Vibrance), revealed significant pesticide potential in inhibiting the pathogen *F. acutatum* (B5, Figure 1). In addition, three of those preparations were highly effective against *F. oxysporum* f. sp. *cepae* (B14, Figure 2), while the fourth compound, Fludioxonil + Sedaxen (Vibrance), was less effective than Azoxystrobin (Amistar).

### 3.2. Seed Pathogenicity Assay

In a follow-up experiment, selected anti-fungal compounds’ efficiency to protect the plants in early growth stages was studied. The selected fungicides-Azoxystrobin + Tebuconazole (Azimut), Prochloraz (Sportak), and Fludioxonil + Sedaxen (Vibrance), were applied in seed coating. These commercial mixtures eliminated almost entirely the growth of the two pathogens inspected, even at the lower concentration tested (1 ppm, Figure 3). Therefore, these compounds were used in seed coating to protect onion seeds against *F. acutatum* (isolate B5) and *F. oxysporum* f. sp. *cepae* (isolate B14) in the Riverside (Orlando) and Noam onion genotypes. In the white Riverside cv., inoculation with *F. acutatum* (B5 isolate) caused a significant delay (approximately 60%, *p* < 0.05) in seed germination (Figure 4A). The addition of the three fungicides in this cultivar caused a significant delay in seed germination (to about 25% and below) and did not help prevent the inhibitory effect of the *F. acutatum* pathogen. In contrast, infection of the seeds with *F. oxysporum* f. sp. *cepae* (B14 isolate) had no measurable influence on Riverside cv. seeds germination. In the red Noam cv., seed germination was not affected by pathogen infection or pesticide toxicity, except for the Fludioxonil + Sedaxen preparation, which caused a 40% delay in germination (*p* < 0.05, Figure 4B). Moreover, in this preparation, in the Noam cv., inoculation with *F. acutatum* did not cause a measurable difference and resulted in a similar effect on seed germination.

The effect of the treatments was noticeable in the shoot length measurements. In white Riverside cv. (Orlando) seeds (Figure 5A), the stalk length was dramatically affected by the chemical treatments and fungal infection (*p* < 0.05). These treatments almost abolished the epicotyl elongation of the germinating seeds. Some measurable (but still minor) epicotyl growth was recorded in *F. oxysporum* f. sp. *cepae* (isolate B14) infected seeds (without the addition of fungicides). In contrast, in the red Noam genotype (which was less affected by the fungicides’ toxicity), Prochloraz was the only chemical with no apparent decreasing effect on shoot development (Figure 5B). This treatment succeeds in preventing the suppression effect of the two *Fusarium* species on epicotyl growth. Indeed, under the infection of *F. acutatum* (isolate B5), the sprouts developed only slightly, and severe early growth suppression was observed. The *F. oxysporum* f. sp. *cepae* (isolate B14) infection was less severe but also significantly (*p* < 0.05) disrupted normal shoot growth. The other two anti-fungal treatments inspected here, Azoxystrobin + Tebuconazole (Azimut) and Fludioxonil + Sedaxen (Vibrance), drastically (*p* < 0.05) suppressed the seeds’ epicotyl development. Regardless of this negative effect, the Fludioxonil + Sedaxen mixture applied to protect the seedlings from the *F. oxysporum* f. sp. *cepae* achieved significantly better shoot development compared to the application of the preparation without the pathogen. This suggests that the mixture may have some protective influence against this pathogen.

As expected, there is some correlation between the shoot length results and the fresh biomass measurement results (Figure 6). At the wet biomass measure, the Riverside cv. was highly sensitive to the fungal inoculation and the fungicide treatments (*p* < 0.05). Despite the high concentration at which they were applied, the inspected fungicides failed to prevent the pathogen’s inhibitory effect on the Riverside cv. sprouts’ initial growth. Here, too, the Noam cv. was less affected by the chemical treatment phytotoxicity. Prochloraz was shown to be an especially promising potential treatment since it had no apparent inhibitory influence on the seedlings’ biomass and exhibited significant (*p* < 0.05) protection against both pathogens species inspected here. Therefore, the Prochloraz preparation was chosen for the subsequent pots experiment.

The in vitro seeds pathogenicity assay images (Figure 7) support the quantitative measurements described above (Figure 4, Figure 5 and Figure 6). In the non-infected controls (Riverside and Noam cultivars), the etiolated sprouts had healthy germination and early development. The infection with *F. acutatum* (B5) or *F. oxysporum* f. sp. *cepae* (B14) led to a severe reduction in the initial growth. Interestingly, the infection in both cultivars is also characterized by typical fungal dense aerial hyphae growth above and adjacent to the seeds. Except for the Prochloraz in the Noam cv., the commercial compounds tested here had a phytotoxic effect at the concentrations tested. The Prochloraz was also excelled in preventing the two Fusarium species’ harmful influence in the Noam cv. The Fludioxonil + Sedaxen mixture was also beneficial in preventing some of the *F. oxysporum* f. sp. *cepae* (B14) symptoms in the Noam cv. (Figure 7).

### 3.3. Onion Seedlings Inoculation Assay

At this final stage of indoor anti-fungal compounds’ selection, the most promising commercial preparation, Prochloraz, was applied to Noam cv. seedlings in pots in a growing room (Figure 8 and Figure 9). In 30-day-old sprouts, this compound had no significant inhibitory effect on the plants’ height or wet biomass. Infecting the seedlings with either one of the *Fusarium* species significantly (*p* < 0.05) reduced the plants’ shoot length and wet biomass (except for *F. acutatum* that did not alter the shoot height). Interestingly, *F. oxysporum* f. sp. *cepae* (isolate B14), which was less virulent in the in vitro seeds’ pathogenicity assay, was more aggressive in the sprout experiment. The addition of Prochloraz to the infected sprouts had no apparent influence on shoot length but had a significant (*p* < 0.05) positive impact on the plants’ wet biomass.

## 4. Discussion

Today, the measures applied in Israel against FBR are few and include a four-year growing cycle and soil disinfection with Metam sodium. At the same time, agricultural contaminated equipment (such as plowshares, etc.) and workers unintentionally allow the disease to continue to spread to new growing areas.

The present study was designed to examine the potential of chemical fungicides to reduce this disease’s damage. Effective substances against the pathogens involved were identified in vitro. Selected preparations were tested in seeds and potted sprouts under controlled conditions. Out of the 10 fungicides tested, the most effective and the least phytotoxic preparation, Prochloraz (commercial name Sportak, Bayer CropScience, Germany, supplied by Gadot Agro, Israel), showed significant results in Noam cv. against one of the leading causes of FBR, the pathogen *F. oxysporum* f. sp. *cepae*. However, it was not effective in the Riverside variety (Orlando cv.) or against the other FBR pathogen inspected, *F. acutatum*.

It was recently reported that in northeastern Israel, several *Fusarium* species could cause FBR, and that their distribution and prevalence are varied among different geographical regions and onion genotypes [3]. Thus, an effective control strategy must take into consideration all the potential risk fungi involved and direct the different solutions to eliminate each of the FBR pathogens. As demonstrated here, an effective chemical treatment against one FBR pathogen complex may not eliminate the threat posed by the other *Fusarium* species involved. It is even possible that chemicals restricting one of the FBR complex pathogens will lead to an imbalance in the inter-species relationships of the phytopathogenic fungi involved. The consequence may be that other partners in this complex become more dominant and cause exacerbation of the disease. Even other potential pathogens that do not belong to the *Fusarium* genus can be associated with this disease and should be taken into account. Such interspecies relationships were demonstrated in other host–pathogen interactions [12,13]. Thus, a tailor-made solution for each of the FBR pathogens (and even for specific geographical regions) may be required in a coordinated control program to deal with this devastating risk. Such a solution may require intensive chemical treatment that has several drawbacks in the short and long run. In the short run, intensive chemical treatment may lead to the appearance of fungicide resistance. Such cases become more and more widespread [14]. Indeed, Prochloraz-responsive genes facilitating DMI-resistance have already been reported in other phytopathogens (see, for example [15,16]. One solution to cope with this problem is the use of a mixture of fungicides, each of which has a different action mechanism (see, for example [17,18]). Another approach is to decrease the fungicide dosages required by combining the chemical treatment with eco-friendly biological solutions [19]. Such an approach was already demonstrated as effective in reducing FBR damages [20,21] and is also very beneficial in coping with the long-term problems of intensive chemical pesticides’ use. In the long-term, chemical control of phytoparasitic fungi may cause environmental, animal, and human hazard risks. Reducing the use of chemical fungicides has become increasingly essential and is nowadays a worldwide effort [19].

It is essential to continue to locate additional fungicides while evaluating their toxicity to the plant and their effectiveness against each of the pathogens involved in causing the disease. Alongside this effort, the development of fast and effective screening methods to evaluate the potential of these pesticides is essential. We should keep in mind that the early tests on growth media plates aim at rapidly screening many fungicides and indicate their effectiveness with minimal investment in time and effort. These will be tested later in seeds and sprouts while reducing inadequate preparations. Only in the final stage, few and highly potential selected chemical preparations will be tested in a field condition experiment throughout a full growing season. While this method is essential to reduce the investment and the lengthy time involved in field experiments, it is not without flaws. It is important to remember that screening in culture plates and seeds, and even potted plants, under controlled conditions has a limited ability to predict field efficacy [22]. Nevertheless, potted sprouts experiments for selected fungicides are still essential. These experiments do not depend on specific climatic conditions required for field growth sessions and are not affected by the high variability in the environmental conditions that accompany open-air experiments. Such indoor sprout experiments are also required to assess anti-fungal compounds’ effectiveness in preventing the initial pathogen penetration into the host onion plants.

The seeds and potted plant experiments presented in this study constitute a set of tools for detecting and testing fungicides against the *Fusarium* species involved in FBR disease. These tests may also indicate the degree of virulence of the pathogens involved and the degree of sensitivity/tolerance of the onion varieties tested. The results obtained are encouraging news for Noam cv. growers in Israel. However, before the final recommendations are given to growers, the results should be established, expanded, and deepened in subsequent studies that will be conducted under field conditions. Prochloraz (Sportak 45%), an ergosterol biosynthesis inhibitor, was found in previous studies [20,23] to be the most effective onion seed fungicide for controlling FBR. In the future, it would be worthwhile to try preparations with active ingredients from the same family (Imidazoles) of Prochloraz and other groups having the potential to delay the development of FBR pathogens. Continued efforts to locate and implement fungicides are essential for improving our ability to deal with FBR disease, which poses an increasing risk to the onion industry.

## 5. Conclusions

Over the past decade, reports have accumulated from farmers about an increase in cases and the spread of *Fusarium* onion basal rot disease (FBR) in fields in Israel. Recent reports indicate increasing concern in this regard in onion fields in the northern part of the country. Species of *Fusarium*, mainly *F. oxysporum* f. sp. *cepae*, are the causes of the disease. Plates screening of fungicides merged in the PDA substrate, in vitro seed pathogenicity assay, and potted seedlings in a growing room revealed that Prochloraz (brand name Sportak, Bayer CropScience, Germany, supplied by Gadot Agro, Israel) has significant pest control potential in delaying the disease in Noam cv. The method presented in this work is essential for scanning multiple compounds and gradually ruling out ineffective ones.

Any pest control plan must address the plant variety and its degree of resistance, growing conditions, and the microflora that sets it apart and is involved in causing the disease. In the case of FBR disease, which is caused by several known virulent *Fusarium* pathogens, the relationship fabric of those species and the possible involvement of other (yet to be identified) *Fusarium* species should be carefully considered. Since onion cultivars show variable resistance, both to the phytotoxicity of the pesticides and the pathogens involved in the disease, it is important to plan a solution that addresses these aspects. An effective protective suit should be tailored to the onion variety and each of the fungal species involved. It is also essential to carefully consider the method of applying the various substances and try to combine fungicides having a different mechanism of action to prevent the development of resistance against the preparation.

## Figures and Tables

**Figure 1 jof-07-00235-f001:**
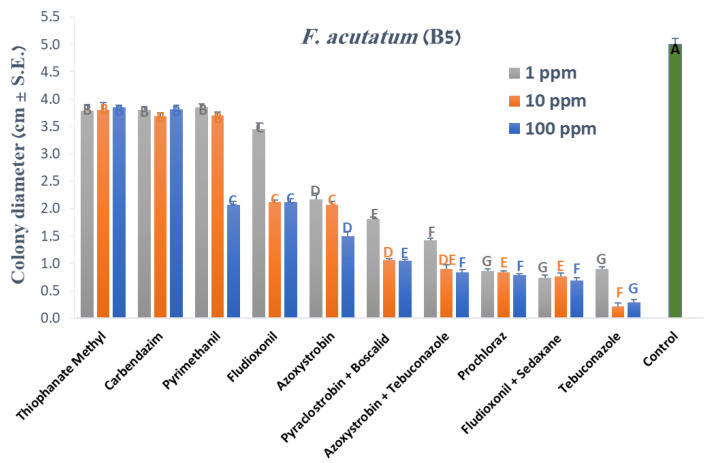
Effect of selected fungicides on the development of *Fusarium acutatum* (isolate B5) in culture media. The fungicides were tested at concentrations of 1, 10 and 100 PPM. The fungicides are described in Table 2. Culture agar discs from a five-day-old colony were grown on Petri dishes containing anti-fungal commercial preparations and incubated under dark conditions at 28 ± 1 °C for four days. The control treatment is colonies that were grown under the same conditions without fungicide. The columns represent an average diameter of six replications, error lines represent standard error. Different letters above the columns represent a significant difference (*p* < 0.05) in the one-way analysis of variances (ANOVA) test.

**Figure 2 jof-07-00235-f002:**
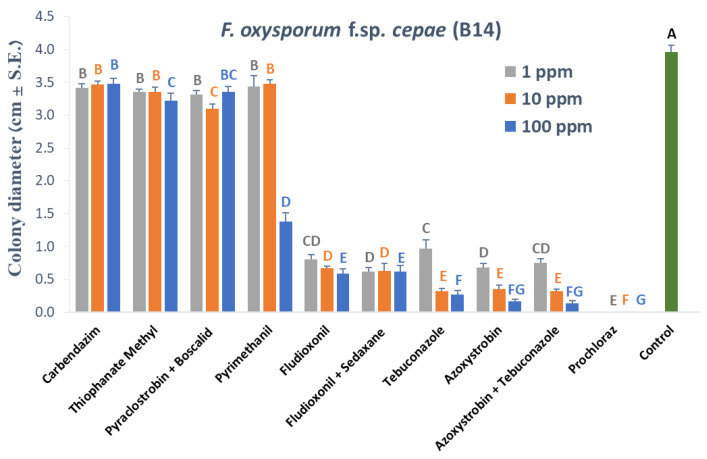
Effect of selected fungicides on the development of *Fusarium oxysporum* f. sp. *cepae* (isolate B14) in culture media. The experiment was conducted as described in Figure 1. The columns represent an average diameter of six replications, error lines represent a standard error. Different letters above the columns represent a significant difference (*p* < 0.05) in the ANOVA test.

**Figure 3 jof-07-00235-f003:**
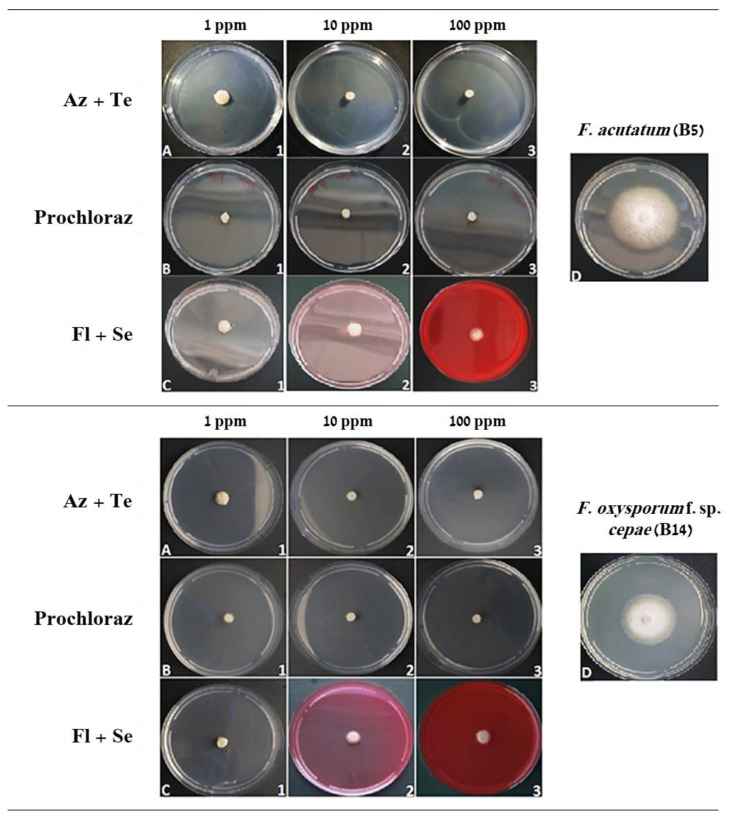
Photograph of the plate assay for assessing the fungicides’ effectiveness against the onion basal rot disease causal agents. The experiment was conducted as described in Figure 1. Quantitative results of the plate assay are presented in Figure 1 and Figure 2. Representative plates of efficient treatments that were chosen for the subsequent evaluation in seed assay are shown. The fungicide treatments are: A. Azoxystrobin + Tebuconazole (Az + Te, Azimut); B. Prochloraz (Sportak); and C. Fludioxonil + Sedaxen (Fl + Se, Vibrance). The control treatment plates (D) were prepared with a PDA substrate (without the addition of fungicide). The fungicides were tested at concentrations of 1, 10 and 100 PPM (1–3, respectively). The dishes were photographed after three days of incubation under dark conditions at 28 ± 1°C.

**Figure 4 jof-07-00235-f004:**
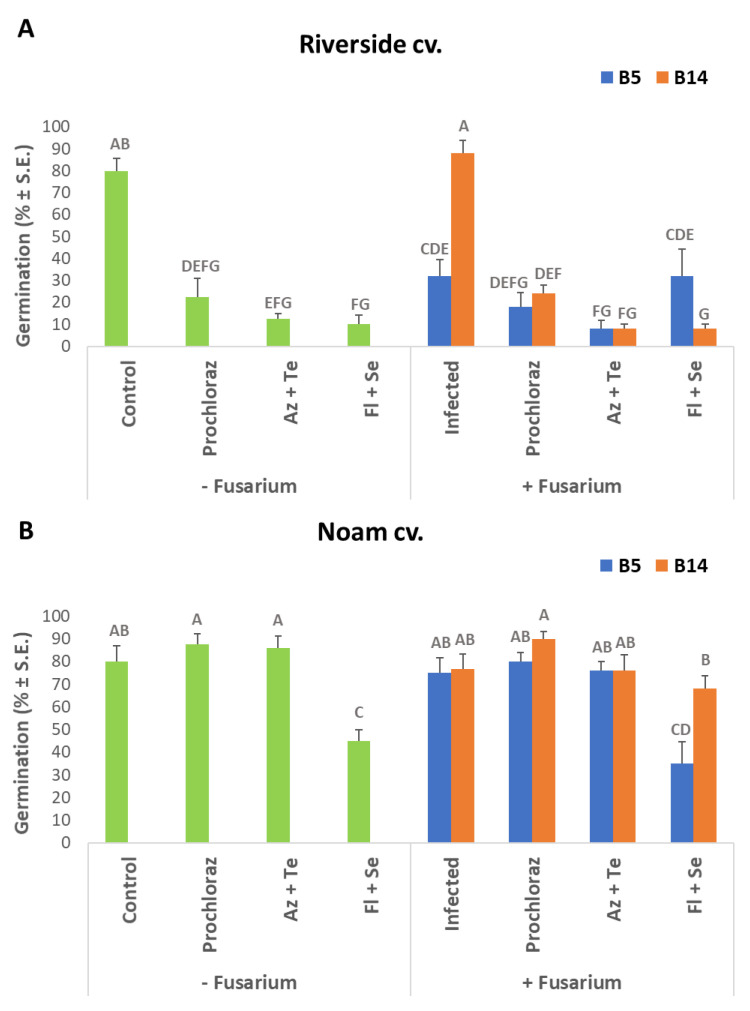
Percentage of Riverside cv. (**A**, Orlando) and Noam cv. (**B**) onion seeds germinated under the influence of *Fusarium* spp., with or without fungicides seed coating. The pathogens *Fusarium acutatum* (isolate B5) and *Fusarium oxysporum* f. sp. *cepae* (isolate B14) were tested separately in this assay. Seeds were coated with Prochloraz (Sportak), Azoxystrobin + Tebuconazole (Az + Te, Azimut), and Fludioxonil + Sedaxen (Fl + Se, Vibrance) fungicides at concentrations of 0.3%, 33%, and 20% (*w*/*w*), respectively. The controls were uninfected seeds and pesticide-free treatments. Seeds were incubated in dark conditions at 28 ± 1 °C for nine days. The columns represent an average of five repetitions, deviation lines represent a standard error. Different letters above the columns represent a significant difference (*p* < 0.05) in the ANOVA test.

**Figure 5 jof-07-00235-f005:**
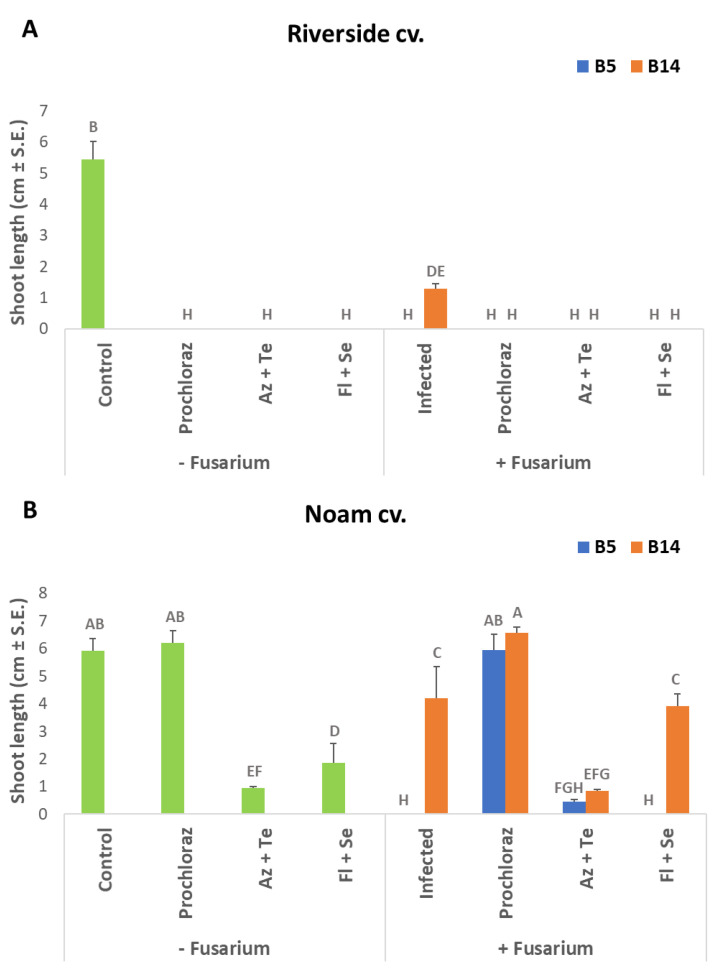
The stalk length of the onion genotypes Riverside (**A**, Orlando) and Noam (**B**) under the influence of selected fungicides and *Fusarium* spp. inoculation. The experiment description and abbreviations are detailed in Figure 4. The columns represent an average of five repetitions, deviation lines represent a standard error. Different letters above the columns represent a significant difference (*p* < 0.05) in the ANOVA test.

**Figure 6 jof-07-00235-f006:**
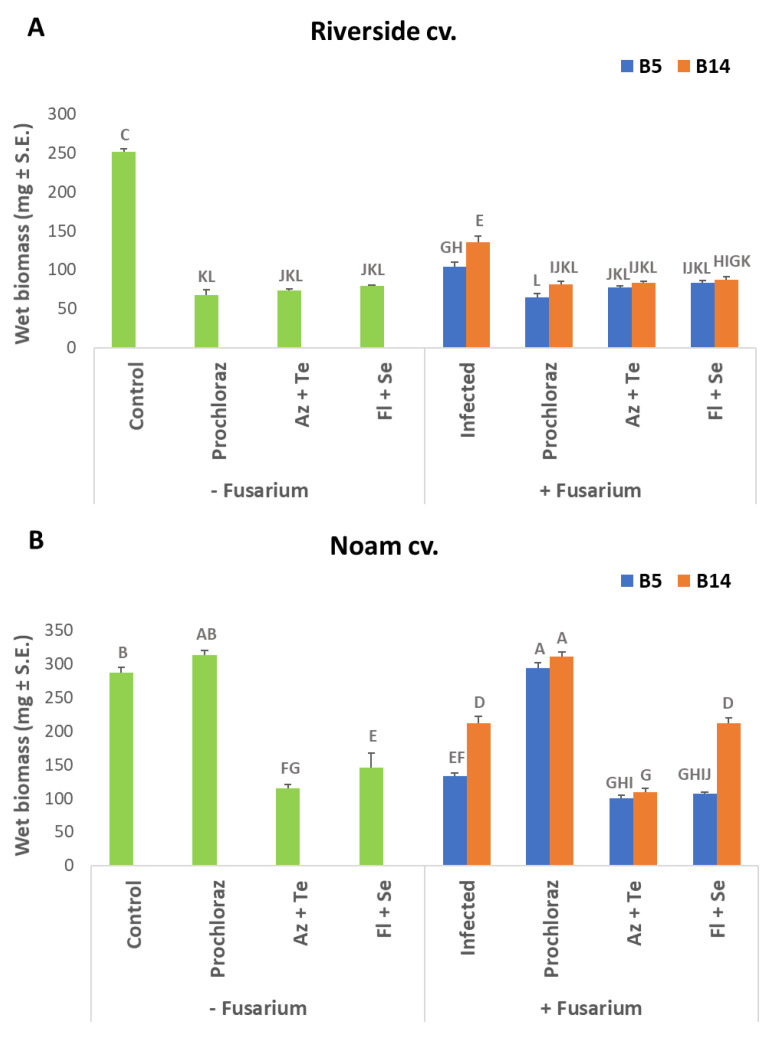
Average fresh biomass of Riverside and Noam genotypes onion seeds after *Fusarium* spp. inoculation and selected pesticide seed coating. The experiment and abbreviations are depicted in Figure 4. The columns represent an average of five repetitions, deviation lines represent a standard error. Different letters above the columns represent a significant difference (*p* < 0.05) in the ANOVA test.

**Figure 7 jof-07-00235-f007:**
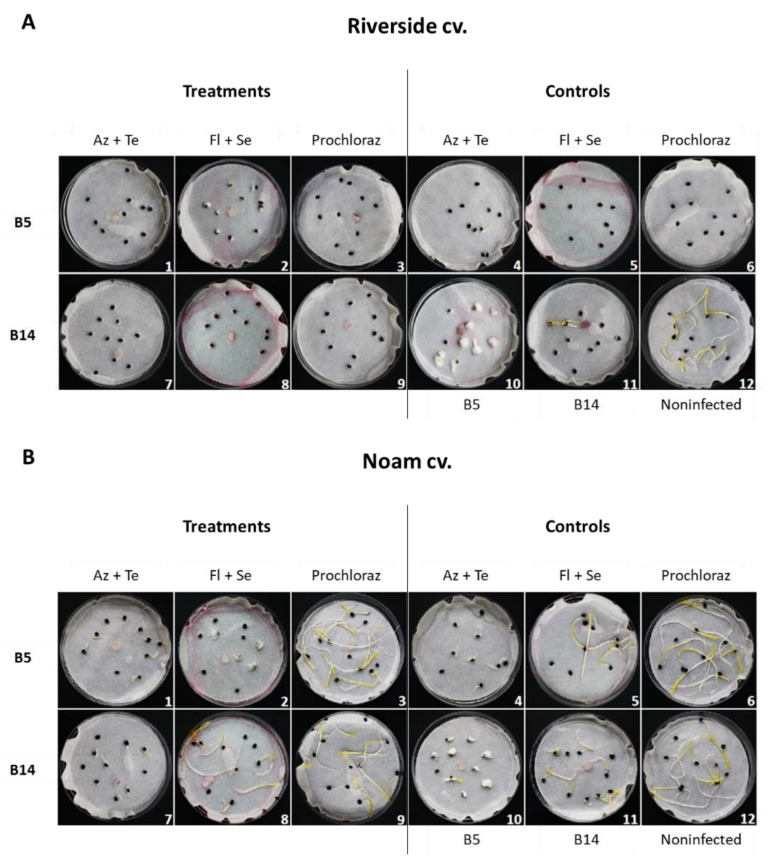
Image of the seeds’ pathogenicity assay for the Riverside cv. (**A**, Orlando) and the Noam cv. (**B**). The experiment and abbreviations are depicted in Figure 4. Treatments: A combination of phytopathogens infection and chemical seed coating (1–3—*Fusarium acutatum*, isolate B5; 7–9—*Fusarium oxysporum* f. sp. *cepae*, isolate B14). Controls: Seeds without chemical coating and pesticide (12, Noninfected), non-inoculated seeds with selected fungicides (4–6), and inoculated seeds without pesticide (10–11).

**Figure 8 jof-07-00235-f008:**
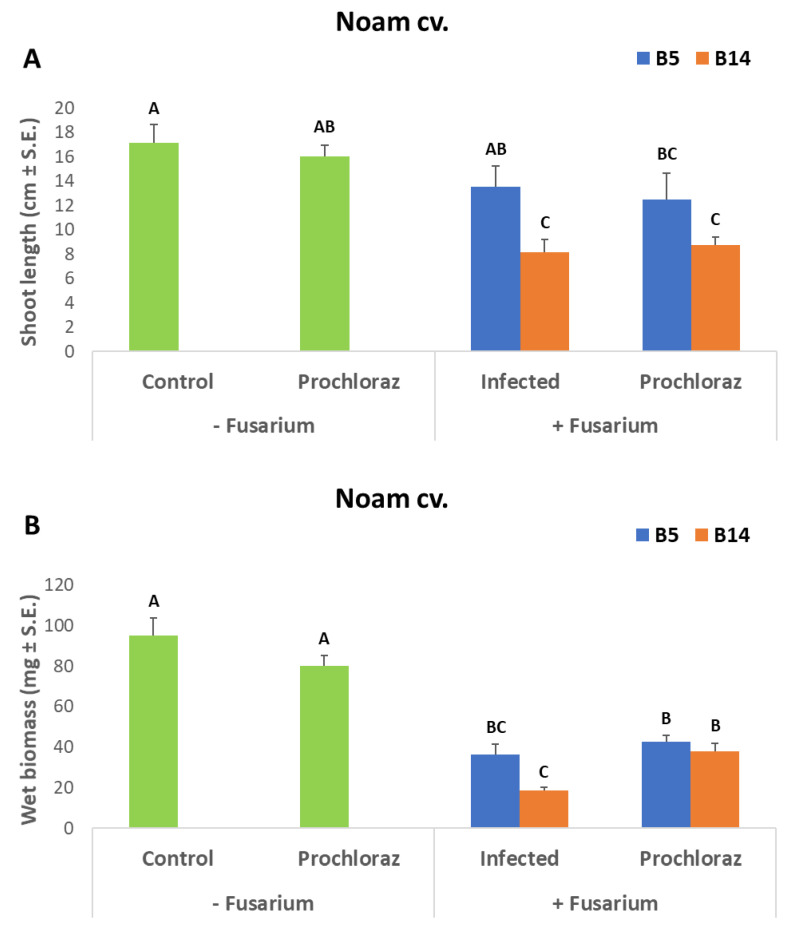
Potted sprouts assay to evaluate the effectiveness of the Prochloraz preparation against *Fusarium* spp. involved in onion basal rot disease. (**A**). Shoot length (**B**). Wet weight. The plants were inoculated separately with pathogens *F. acutatum* (isolate B5) and *F. oxysporum* f. sp. *cepae* (isolate B14). The Prochloraz (at a concentration of 0.3% *w*/*w*) treatment was performed 14 days after sowing by irrigation with 100 mL tap water per pot. Controls: Untreated and non-inoculated sprouts, non-inoculated sprouts treated in Prochloraz, and infected plants without Prochloraz. Values were measured after 30 days in a growing room with artificial light for 16 h and eight hours in darkness, humidity percentage of 45–50%, and a temperature of 28 ± 3 °C. The columns represent an average of five repetitions, deviation lines represent a standard error. Different letters above the columns represent a significant difference (*p* < 0.05) in the ANOVA test.

**Figure 9 jof-07-00235-f009:**
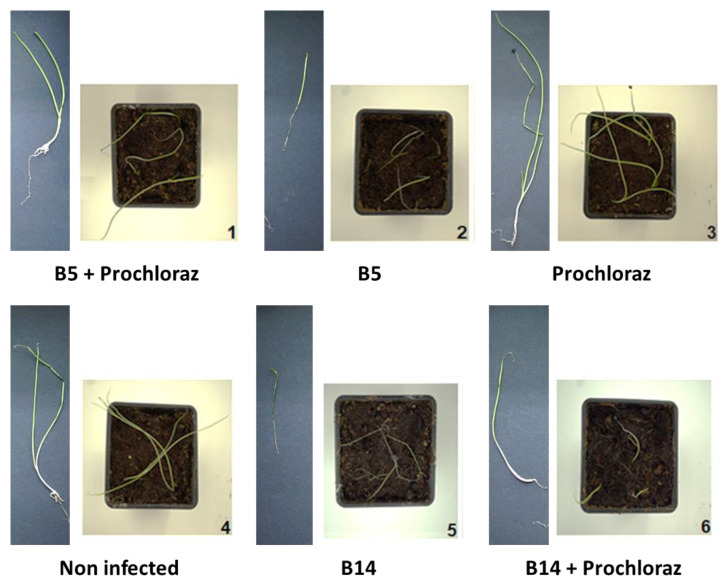
Photograph of representative pots (**right**) and sprouts (**left**) from the assay to evaluate the effectiveness of Prochloraz on emerging seedlings. The experimental procedure and the plants’ growth parameters results are described in Figure 8.

**Table 1 jof-07-00235-t001:** *Fusarium* isolates used in this study ^a^.

Species	Isolate	NCBI Accession and Score	Collection Sites ^b^	Onion Cultivar ^c^
*F. acutatum*	B5	MK507814.1 (100%)	Kibbutz Ortal	Riverside
*F. oxysporum* f. sp. *cepae*	B14	KP964881.1 (99.55%)	Moshav Eliad	565/505

^a^ Strain B5 was isolated on 30 August 2017, while Strain B14 was isolated on 23 May 2018. ^b^ Kibbutz Ortal is located in the northern Golan Heights; Moshav Eliad is situated in the southern Golan Heights [3]. ^c^ Riverside (Orlando) cv.—white onion. 565/505 cv.—newly developed red onion. Both are supplied by Hazera Seeds Ltd., Berurim M.P. Shikmim, Israel.

**Table 2 jof-07-00235-t002:** Fungicides used in this study ^a^.

Fungicide Commercial Name	Manufacturer Supplier	Active Ingredient (Common Name and Chemical Structure)	Group Name	Chemical Group	Target Site of Action	Section Mode of Action	Active Ingredient (g/l)
Sportak	Bayer CropScience (Monheim am Rhein, Germany) Gadot Agro (Kidron, Israel)	Prochloraz 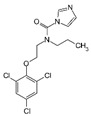	DMI-fungicides (demethylation inhibitors)DMI-fungicides (demethylation inhibitors(SBI: Class I)	Imidazoles	C14- demethylase in sterol biosynthesis (erg11/cyp51)	Sterol biosynthesis in membranes	450
Celest 100FS	Syngenta(Basel, Switzerland), Gadot Agro (Kidron, Israel)	Fludioxonil 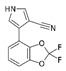	PP-fungicides (phenyl pyrroles)	Phenylpyrroles	MAP/histidine kinase in osmoticsignal transduction(os-2, HOG1)	Signal transduction	100
Signum W.G.(Bc + Ps)	BASF(Ludwigshafen,Germany),Agan(Ashdod, Israel)	Boscalid 26.7% 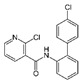	SDHI (succinate dehydrogenase inhibitors)	Pyridine-carboxamides	Succinate dehydro-genase	Respiration	267
Pyraclostrobin 6.7% 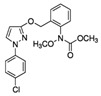	QoI-fungicides(quinone outside inhibitors)	Methoxy-carbamates	Cytochrome bc1(ubiquinoloxidase) at Qosite (*cyt b gene)*	67
Topaz W.P.	Nippon Soda (Japan), Adama Makhteshim(Be’erSheva, Israel)	Thiophanate-Methyl 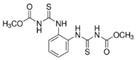	Fungicides (methyl benzimidazole carbamates)	Thiophanates	ß-tubulin assembly in mitosis	Cytoskeleton and motor protein	700
Mythos 300 SC	Buyer (Cyprus)Lidor Chemicals (Israel)	Pyrimethanil 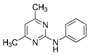	AP-fungicides (anilino pyrimidines)	Anilino-pyrimidines	Methionine biosynthesis (proposed) (cgs gene)	Amino acids and protein synthesis	300
Orius 25	Adama Irvita (Netherlands) Adama Makhteshim(Be’erSheva, Israel)	Tebuconazole 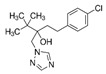	DMI-fungicides (demethylation inhibitors) (SBI: Class I)	Triazoles	C14- demethylase in sterol biosynthesis (erg11/cyp51)	Sterol biosynthesis in membranes	250
Delsene	Agro-Chemie (Hungary) Gadot Agro (Kidron, Israel)	Carbendazim 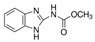	MBC-fungicides (methyl benzimidazole carbamates)	Benzimidazoles	ß-tubulin assembly in mitosis	Cytoskeleton and motor protein	500
Azimut	Adama Makhteshim(Be’erSheva, Israel)	Azoxystrobin 12% 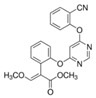	QoI-fungicides(quinone outside inhibitors)	Methoxy-acrylates	Respiration C3:cytochrome bc1(ubiquinol oxidase) at Qo site (cyt b gene)	Respiration	120
Tebuconazole 20% 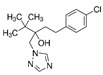	DMI-fungicides (demethylation inhibitors) (SBI: Class I)	Triazoles	C14- demethylase in sterol biosynthesis (erg11/cyp51)	Sterol biosynthesis in membranes	200
Amistar	Syngenta (Basel, Switzerland), Adama Makhteshim (Airport City, Israel)	Azoxystrobin 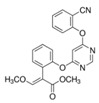	QoI-fungicides(quinone outside inhibitors)	Methoxy-acrylates	Respiration C3:cytochrome bc1(ubiquinol oxidase) at Qo site (cyt b gene)	Respiration	250
Vibrance	Syngenta(Basel, Switzerland), Gadot Agro (Kidron, Israel)	Fludioxonil 2.5% 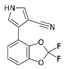	PP-fungicides (phenyl pyrroles)	Phenylpyrroles	MAP/histidine kinase in osmoticsignal transduction(os-2, HOG1)	Signal transduction	25
Sedaxane 2.5% 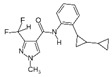	SDHI(succinate dehydrogenaseinhibitors)	Pyrazole-4- carboxamides	Complex II: succinate-dehydrogenase	Respiration	25

## Data Availability

The datasets generated during and/or analyzed during the current study are available from the corresponding author on reasonable request.

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
