# Peer review of "Assessment of Commercial Fungicides against Onion (Allium cepa) Basal Rot Disease Caused by Fusarium oxysporum f. sp. cepae and Fusarium acutatum"

_jof, 2021, doi:10.3390/jof7030235_

Round 1

Reviewer 1 Report

Line 17: i found that you used pesticides in this line and the rest of the work, but shall it be more specific with fungicide.  

Line 24,25 : Reformulate the sentence because it is not clear and connect them  

Line 45 : I found that you used the term"Infested" while it should be Infected" since the work was on microorganisms and not on animals or arthropods,try to adjust it in the rest of the work.  

Line 60-64 : You should insert a Reference.  

Line 71:How there is a gap in identifying the pathogens if you already said before that already been isolated 4 species.  

Line 170: you have to provide some photos for the symptoms.  

Line 180: It should be "Plate assay" instead of "seeds pathogenicity assay.   first reference contain a wrong word Technology.

Author Response

Responses to Reviewer’s 1 Comments

We thank the reviewer for investing substantial efforts, which are undoubtedly contributing to this manuscript. The remarks and suggestions improved this paper’s scientific soundness and accurateness. Your contribution is greatly appreciated.

Line 17: I found that you used pesticides in this line and the rest of the work, but shall it be more specific with fungicide.

The reviewer is correct; although both terms are common in such reports, the term “fungicide” more accurately described the experimental procedures and results. We, therefore, replaced the term “pesticides” with the term “fungicide” in eight places throughout the text.

Line 24-25: Reformulate the sentence because it is not clear and connect them.

The reviewer is right:

The previous version of the sentence: “The most promising compound, Prochloraz (0.3% w/w concentration), had no toxicity influence on the Noam cv. sprouts’ germination and early development, and positive efficacy in reducing the disease’s early growth inhibition symptoms.”,

was replaced by the following version (lines 18-21): “Prochloraz (0.3% w/w concentration), the most promising compound, was efficient in reducing the Noam cv. sprouts’ disease symptoms. This preparation had no harmful toxicity effect and did not influence the plants’ seed germination and early development.”

Line 45: I found that you used the term “Infested” while it should be Infected” since the work was on microorganisms and not on animals or arthropods, try to adjust it in the rest of the work.

As suggested by experts in phytopathology, the term “infested” usually refers to naturally infested soil or fields. The word “infected” refers to the deliberate inoculation of soils. If you disagree with this, we can easily alter all the terms to “infected.”

Line 60-64: You should insert a Reference.

A reference was added as suggested.

Line 71: How there is a gap in identifying the pathogens if you already said before that already been isolated 4 species.

We agree, this sentence should be rewritten and clarified. Thus, we made the following changes:

The previous version of the paragraph: “Still, significant knowledge gaps exist in Israel regarding the disease’s nature and distribution, the pathogens involved in causing it, and the control methods applied against it.”,

was replaced by the following paragraph (lines 65-67): “Still, other pathogenic Fusarium species may be involved in FBR, and significant knowledge gaps exist in Israel regarding the disease’s nature and distribution and the control methods applied against it.

Line 170: you have to provide some photos of the symptoms.

A plate assay photo was added as suggested. See new Figure 3. The text was updated accordingly (lines 215-217): “These commercial mixtures eliminated almost entirely the growth of the two pathogens inspected, even at the lower concentration tested (1 ppm, Figure 3).”

Line 180: It should be “Plate assay” instead of “seeds pathogenicity assay.”  

True, this is a typo. It was corrected as advised.

First reference contains the wrong word Technology.

The reference was checked, and it is now correct.

Reviewer 2 Report

Concerning the revision of the manuscript titled " Inspection of chemical treatment against onion (Allium cepa) basal rot disease caused by Fusarium oxysporum f. sp. cepae and Fusarium acutatum.. I see the point of the study should be clear; it is an evaluation of some fungicides or is a kind of method to testing the available fungicide that existed in the market. Also, the study needs an open field experiment to make the selection decision of one or two of the examined fungicide could be applied. Also, the missing figures of the pots experiment and petri dishes are needed. The constituents of the soil or the soil analysis used in the study should be listed.  Alos, many comments are listed in enclosed manuscript.

Author Response

Responses to Reviewer’s 2 Comments

We would like to express our sincere appreciation to the reviewer for essential and helpful advice. The time and effort invested are greatly appreciated and certainly contributed to the manuscript and improved it. Thank you.

Concerning the revision of the manuscript titled “Inspection of chemical treatment against onion (Allium cepa) basal rot disease caused by Fusarium oxysporum f. sp. cepae and Fusarium acutatum.. I see the point of the study should be clear; it is an evaluation of some fungicides or is a kind of method to testing the available fungicide that existed in the market.

This remark is true. The research goal should be clearly stated. Thus, the third sentence in the Abstract was corrected to: “Here, commercial chemical fungicides were evaluated as control treatments against two of the primary pathogens involved, F. oxysporum f. sp. cepae and F. Acutatum.” (lines 13-14)

Also, the study needs an open field experiment to make the selection decision of one or two of the examined fungicide could be applied.

We agree that this additional step, the field experiment, will eventually lead to the final decision as to which treatments should be applied on a commercial field scale.

We feel that this future work should be conducted and presented in a separate article. The current manuscript already contains many significant experimental results that should be given to encourage further follow-up research in the field.

A field trial with the selected fungicides is indeed our next near-future goal, but it will require several months to accomplish it, and we think it is worth another article since it will produce many new results.

Combining the current manuscript with a field trial result will lead to a very long and loaded article, therefore, we feel it is better to split the work into two steps and present them separately.

Also, the missing figures of the pots experiment and petri dishes are needed.

A plate assay photo was added as suggested. See new Figure 3. The text was updated accordingly (lines 215-217): “These commercial mixtures eliminated almost entirely the growth of the two pathogens inspected, even at the lower concentration tested (1 ppm, Figure 3).”

A photograph of the pots experiment was added (new Figure 9) as suggested by the reviewers, and the relevant text was updated accordingly (lines 300-302): “At this final stage of indoor anti-fungal compounds’ selection, the most promising commercial preparation, Prochloraz, was applied to Noam cv. seedlings in pots in a growing room (Figures 8-9).”

The constituents of the soil or the soil analysis used in the study should be listed.

The soil parameters used for this study were added to the text (lines 150-153): “In the experiment, 0.5-liter pots were used with a non-sterile commercial garden soil mixture (Garden Mix, Deshanit, Be’er Yaakov, Israel) composed of coconut peat, fibers, a relatively low amount of tuff, and Osmocote (ScottsMiracle-Gro, Marysville, Ohio, United States), a 3-4 month slow-release fertilizer.”

Respond to the comments listed in the enclosed manuscript:

Page 1

Title: “inspection” – should be changed into Evaluation or assessment.

The word “inspection” in the title was replaced with “assessment,” as advised.

Line 16: please you have to have a macroscopic image for the affected fungal hyphae. This will give a good explanation for the efficacy of the fungicide.

Indeed we agree. As suggested, a plate assay photo was added (see detailed explanation above).

Line 18: in situ-toxicity.

Corrected as advised.

Line 23: which you used in seed treatment is not kind of control, it is a protective manner. So, please change this sentence under your convey.

The word “control” was replaced with “protective” as suggested.

Line 24: if your study concerning a method for testing the activity of the available fungicides in the market, why you did not indicate for that in the manuscript title.

We agree. The manuscript title was rewritten and now appears as: “Assessment of commercial fungicides against onion (Allium cepa) basal rot disease caused by Fusarium oxysporum f. sp. cepae and Fusarium acutatum.

And is that one method or different methods? Please be sure.

Corrected to “methods”.

Line 26: pot experiment is not enough to evaluate a specific activity for specific fungicide, you must apply this work in the open filed under non controlled conditions, to get a real results.

This is true. As detailed in our response above, the current work presents the methods and encouraging results obtained in a relatively fast experiment series. These will allow a wise choice of successful anti-fungal preparations for a field experiment, which will be performed later. We believe that the stages presented in the current work are crucial for ruling out ineffective compounds and focusing the subsequent field experiment (that requires considerable investment in time, labor, and resources) on the most promising fungicides. The current work is already presenting a substantial amount of experimental results. These findings now encourage testing these preparations in a follow-up experiment to be conducted in the field.

Page 2

Line 47: please add references here along the appearance of this disease in your cultivated areas, and the exact economic losses that resulted in this disease incidence in your country.

Two references were added as suggested to support the information presented (lines 47-48): “For many years, FBR has been known to be one of the most harmful diseases in Israel’s onion cultivation [3,4].”

The exact economic losses that resulted in this disease incidence in Israel is already presented a few lines below (lines 53-59): “The exact loss of marketable bulbs in Israel due to the disease is unknown. The extent of infected crops in the growing area can reach 1%, but losses can extend beyond the field (data according to the Israeli Ministry of Agriculture and Rural Development). Infected onions do not always show disease symptoms, and if they arrive at storage facilities, the problem is considerably worsened [3]. When stored in open sheds or packing houses, the disease can quickly spread to other onion bulbs.”

Line 54: reference here, to ensure what you postulate.

A reference was added as requested (lines 54-56): “The extent of infected crops in the growing area can reach 1%, but losses can extend beyond the field (data according to the Israeli Ministry of Agriculture and Rural Development).”

Lines 82-83: I see that the authors always remember Turkey and India in the study as inspector for Israel, I see that is not good and considered as a defect in the manuscript.

Turkey and India are the world’s largest onion producers. They are therefore given as an example of the damages of the onion disease and their economic consequences. The disease in those and other leading onion-growing countries has a severe global economic impact, therefore, finding preventative treatments is urgently needed.

Page 3

Line 120 (Table 1): Please add a new column contains the chemical structure of each compound.

The chemical structure was added to each of the compounds as suggested.

Page 4

Line 121: The head title should be seed pathogenicity assay and seed germination.

The Section 2.3 title was corrected to “Seed germination pathogenicity assay.”

Line 149: The soil should be sterilized by autoclaving and where from this soil were obtained. what is the soil mixture consists of? and how they collected?

The soil details were added to the text (lines 150-154): “In the experiment, 0.5-liter pots were used with a non-sterile commercial garden soil mixture (Garden Mix, Deshanit, Be’er Yaakov, Israel) composed of coconut peat, fibers, a relatively low amount of tuff and Osmocote (ScottsMiracle-Gro, Marysville, Ohio, United States), a 3-4 month slow-release fertilizer.”

Sterilization of the soil by autoclaving is better suited to field soils that are naturally infested. The autoclaving may alter the soil properties and was not necessary for our experiments since the uninfected control treatment plants were healthy and had normal growth parameters.

Page 5

Line 165: what about the water used in pots irrigation, tap water or sterile water?

The water used for the pots’ irrigation was tap water. This explanation was added to the text (lines 162-166): “About a week after the second inoculation, the anti-fungal compound treatment was carried out by irrigating with 100 ml tap water per pot. The fungicide Prochloraz concentration, 0.3% w/w, was chosen according to the manufacturer’s recommendation. The treatments without the pesticide were watered with 100 ml of tap water.”

Page 6

Line 188: could you please add a figure showing the growth of the fungi against the examined fungicides.

As detailed above, a plate assay photo was added (new Figure 3) as suggested, and the relevant text was updated accordingly.

Page 12

Line 356: more discussion is needed.

The entire Discussion section was reworked, expanded an improved.

Line 357: conclusion should be shorten.

The Conclusion section was shortened as suggested and is now more focused and purposeful.

Page 13

Line 382: also you should add a convey which onion cultivar could be cultivated in your country based on your study, otherwise a new fungicide should be discovered to gave a chance for the two examined cultivars to cultivated.

We agree, thus the flowing sentences were updated in the Discussion and Conclusions sections:

  • Discussion, lines 381-384: “The results obtained are encouraging news for Noam cv. growers in Israel. However, before the final recommendations are given to growers, the results should be established, expanded and deepened in subsequent studies that will be conducted under field conditions.”
  • Conclusions, lines 397-400: “Plates screening of fungicides merged in the PDA substrate in vitro seed pathogenicity assay, and potted seedlings in a growing room revealed that Prochloraz (brand name Sportak, Bayer CropScience, Germany, supplied by Gadot Agro, Israel) has significant pest control potential in delaying the disease in Noam cv.”

This manuscript is a resubmission of an earlier submission. The following is a list of the peer review reports and author responses from that submission.